# Correlative Study on Impaired Prostaglandin E2 Regulation in Epicardial Adipose Tissue and Its Role in Maladaptive Cardiac Remodeling via EPAC2 and ST2 Signaling in Overweight Cardiovascular Disease Subjects

**DOI:** 10.3390/ijms21020520

**Published:** 2020-01-14

**Authors:** Elena Vianello, Elena Dozio, Francesco Bandera, Marco Froldi, Emanuele Micaglio, John Lamont, Lorenza Tacchini, Gerd Schmitz, Massimiliano Marco Corsi Romanelli

**Affiliations:** 1Department of Biomedical Sciences for Health, University of Milan, 20133 Milan, Italy; elena.dozio@unimi.it (E.D.); francesco.bandera@unimi.it (F.B.); lorenza.tacchini@unimi.it (L.T.); mmcorsi@unimi.it (M.M.C.R.); 2Cardiology University Department, Heart Failure Unit, IRCCS Policlinico San Donato, 20097 Milan, Italy; 3Department of Clinical Sciences and Community Health, University of Milan, 20122 Milan, Italy; marco.froldi@unimi.it; 4Internal Medicine Unit IRCCS Policlinico San Donato, San Donato Milanese, 20097 Milan, Italy; 5U.O.C. SMEL-1 of Clinical Pathology, IRCCS Policlinico San Donato, San Donato Milanese, 20097 Milan, Italy; emanuele.micaglio@grupposandonato.it; 6Randox Laboratories LTD, R&D, Crumlin-Antrim, Belfast, BT29, Northen Ireland, UK; 7Department of Clinical Chemistry and Laboratory Medicine, University Hospital Regensburg, 93053 Regensburg, Germany

**Keywords:** epicardial adipose tissue (EAT), prostaglandin E2 (PGE_2_), EP3 receptor, EP4 receptor, exchange protein directly activated by cAMP isoform 2 (EPAC2), stimulating growth factor 2 (ST2), interleukin(IL)-33, Cardiovascular Diseases (CVDs), fat mass

## Abstract

There is recent evidence that the dysfunctional responses of a peculiar visceral fat deposit known as epicardial adipose tissue (EAT) can directly promote cardiac enlargement in the case of obesity. Here, we observed a newer molecular pattern associated with LV dysfunction mediated by prostaglandin E2 (PGE_2_) deregulation in EAT in a cardiovascular disease (CVD) population. A series of 33 overweight CVD males were enrolled and their EAT thickness, LV mass, and volumes were measured by echocardiography. Blood, plasma, EAT, and SAT biopsies were collected for molecular and proteomic assays. Our data show that PGE_2_ biosynthetic enzyme (PTGES-2) correlates with echocardiographic parameters of LV enlargement: LV diameters, LV end diastolic volume, and LV masses. Moreover, PTGES-2 is directly associated with EPAC2 gene (r = 0.70, *p* < 0.0001), known as a molecular inducer of ST2/IL-33 mediators involved in maladaptive heart remodelling. Furthermore, PGE_2_ receptor 3 (PTEGER3) results are downregulated and its expression is inversely associated with ST2/IL-33 expression. Contrarily, PGE_2_ receptor 4 (PTGER4) is upregulated in EAT and directly correlates with ST2 molecular expression. Our data suggest that excessive body fatness can shift the EAT transcriptome to a pro-tissue remodelling profile, may be driven by PGE_2_ deregulation, with consequent promotion of EPAC2 and ST2 signalling.

## 1. Introduction

Perturbations of signaling in the heart and vessels of the body are the leading causes of cardiovascular disorders including coronary artery diseases (CAD) and valve heart diseases (VHD) [1]. Among different stimuli which can contribute to alter cardiac and vessel molecular responses, excessive fat body can be considered one of the primary causes associated with maladaptive heart response [2]. Obese patients are at increased risk of cardiovascular disease (CVD) and heart failure (HF) due to the hemodynamic stresses related to abnormal body mass and increase of volume overload [3]. The excessive hemodynamic stresses lead to cardiac microvascular rarefaction and fibrosis, especially with abnormalities of cardiac diastolic filling [4].

The first effect of chronic volume overload on left ventricle (LV), directly related to fatness, is characterized by typically LV and left atria (LA) dilation, with a preserved ejection fraction (EF), suggesting proportional expansion of plasma volume with body mass. For this reason, the capacity of LV to dilate in response to hydrodynamic volume overload is impaired and disproportionate.

Since overweight is the most deputed cause of these cardiac outcomes, it is assumed that visceral adiposity drives both vessels derangements and heart fibrosis [5], but the molecular patterns relating to these abnormalities are not fully understood. There is recent evidence suggesting that during maladaptive adipose tissue remodeling, dysfunctional fat responses of a particular visceral fat closely surrounding the heart and all arteries- known as epicardial adipose tissue (EAT)-can be a metabolic transducer of both local and systemic inflammation, through the direct release into myocardial microcirculation of bioactive fibro-adipokines [6,7] involved in heart metabolism [8,9,10,11,12,13]. EAT thickness varies 1 mm to a maximum of almost 23 mm [13]. This wide range probably reflects the substantial differences in abdominal visceral fat distribution [13]. Previous studies found median epicardial fat thicknesses of 7 mm in men and 6.5 mm in women in a large population of patients who were examined by transthoracic echocardiography for standard clinical indication [13]. Variations in EAT size can be considered a potential cardiovascular risk factor due to the shift in the production of active cardiometabolic mediators. Among these, one peculiar adipokine known as soluble stimulating growth factor 2 (sST2) [14] has attracted attention on account of both fat mass enlargement and maladaptive heart response due to its role in silencing the main mechanosensitive system [15,16] expressed and activated both in adipose tissue and in the heart [17]; this comprises by two immune mediators, the transmembrane isoform of ST2 (ST2L) and its natural ligand, the interleukin (IL)-33, one of the main alarmin proteins in the body [18]. The IL-33/ST2L system in adipose tissue tries to maintain the size of the fat mass, controlling intracellular nuclear factors that regulate adipocytes number and size of the adipocytes [19,20]. In the heart IL-33/ST2L system has cardioprotective effects since under biomechanical stress it promotes cell survival and prevents apoptosis and fibrosis [21]. However, in pathological conditions such as obesity and cardiovascular disorders, both adipocytes and cardiac cells release larger amount of sST2; this functions as a decoy receptor, sequestering IL-33, losing its cardio-fat protection properties through ST2L binding, and consequently promoting an increase in fat mass, and heart damage [14,16,19,22]. In a previous study [23] we identified as potential molecular inducer of this system in EAT, the exchange proteins directly activated by cAMP (EPACs), which are the main effectors of the second messenger in the body, the cycle adenine monophosphate(cAMP) [23,24]. The EPAC protein family is composed of EPAC1 and EPAC2 which in adipose tissue control adipogenesis and lipolysis and are induced by cAMP [25,26]. We reported that when EAT thickness increases, the local upregulation of EPAC2 promotes an alarm profile associated with maladaptive remodeling in which ST2 gene, encoding for both cardiac stretch proteins, ST2L and sST2, correlated with local expression of EPAC2 [23].

Prostaglandin E2 (PGE_2_) appears to have a crucial role in intracellular cAMP concentration in visceral fat [27,28,29]. PGE_2_ is a potent lipid mediator secreted by various cell types in visceral adipose tissue and it appears to be implicated in the regulation of inflammation and adipocytes functions [30]. Different studies have noted that the excess accumulation of visceral fat depends on depot-specific expression of key enzymes involved in adipose tissue functions, including PGE_2_ biosynthesis-related enzymes [31].

PGE_2_ is the principal prostaglandin produced by visceral adipose tissue including EAT deposit [27], which regulates energy metabolism and, particularly in obesity, contributes to fat-inflammation and obesity-related insulin resistance, through the activation of prostaglandin-endoperoxide synthase 2 (PTGES-2) [32], known also as cyclooxygenase 2. PGE_2_ regulates adipose functions and exerts its biological effects through its four receptors (EP1, EP2, EP3 and EP4) [27]. The EP1 receptor is involved in intracellular Ca^2+^ level. The EP3 receptor is designed to lower intracellular cAMP concentration through the inhibition of adenylyl cyclases (ADCYs), thus promoting adipogenesis [33,34]. EP2 and especially EP4 have opposite effects, stimulating the increase of intracellular cAMP levels with the promotion of lipolysis [35,36,37,38]. The different role of PGE_2_ receptors in the control of intracellular cAMP concentration, led to them being considered the master regulators of adipogenic and lipolytic processes, and their deregulation can be associated with obesity-related disorders. By eliciting signaling through cAMP and its effectors, the EPACs proteins, PGE_2_ can potentiate the expression of mechanosensitive system IL-33/ST2L in immune cells [39].

Due to the importance of ST2 gene both in heart and fat metabolism, we set out to identify a new molecular gap among PGE_2_ metabolism in EAT of overweight CVD persons and the expression of ST2 cardiac stretch mediator via EPAC2 gene, as new potential molecular pattern of maladaptive heart response.

## 2. Results

### 2.1. CVD Patients’ Main Characteristics and Echocardiography

Anthropometric, clinical data and body fatness measurements are set out in Table 1. The CVD subjects had higher indices of body fatness and different in BMI (27.95 ± 5.19 vs normal range 18.50–24.99), waist circumference (106.70 ± 15.08 cm vs normal reference value less than 94 cm) and waist:hip ratio (WHR) (0.98 ± 0.13 normal cut off less than 0.95 ), indicating that CVD patients were overweight. 

Biochemical parameters associated with body fatness were intra-male reference ranges except for N-terminal pro B-type natriuretic peptide (NT-pro-BNP) (453.92 ± 596 pg/mL) and C-reactive protein (CRP) (0.98 ± 0.38 mg/100 mL) which are the clinical parameters currently most used for cardiac stress assessment and body inflammation.

Echocardiography parameters of our CVD subjects are reported in Table 2. EAT thickness was evaluated both in end-diastolic and end-systolic echocardiographic frames and we used the end-systolic frame as EAT measurement because it is considered the best cardiac moment to detect EAT thickness [13,40]. The average EAT was about 7 mm. A normal upper-limit value for EAT thickness has yet not been established. 

Echocardiography showed that overweight CVD subjects typically had left ventricle (LV) and atria (LA) dilation, with a preserved ejection fraction (EF), CVD subjects presented an eccentric LV hypertrophy (relative wall thickness: RTW < 0.42% and indexed LV mass: LVM/BSA ≥115 g/m^2^), the first effect of chronic volume overload on LV, directly related to fatness [7]. 

### 2.2. Anthropometric Measures of Body Fatness Are Associated with Maladaptive Heart Remodeling in Overweight CVD Subjects

The acknowledged parameters of fat body distribution as BMI and waist circumference, directly correlate with echocardiographic indexes of heart maladaptation (Table 3). BMI is not only a predicting factor of insulin resistance due to the positive correlations with Homeostatic Model Assessment for Insulin Resistance (HOMA), fast insulin, waist circumference and an inverse relation with HDL cholesterol. It also directly correlates with both the diameters and volumes of LV, with its mass and LA size. The LA enlargement and dysfunction are the most predictors of (HF) in overweight patients with CVD. The alternative measure that reflects abdominal adiposity waist circumference, which has been suggested as superior to BMI in predicting CVD outcomes, is directly related to EAT thickness and with the indexes predicting insulin resistance. Waist circumference is also related to both LV and LA enlargement. 

### 2.3. Prostaglandin-Endoperoxide Synthase 2 (PTGES-2) Expression in EAT is Directly Related to Maladaptive Heart Remodeling Indexes in Overweight CVD Subjects

In view the importance of body fatness on heart maladaptation, we investigated the PGE_2_ molecular alterations in EAT from overweight CVD patients. Considering the role of PTGES-2 as a mediator of adiposity and its involvement in fat-inflammation and obesity-related disorders, including cardiovascular complications, we ran a correlation analysis between PTGES-2 molecular expression and echocardiographic parameters of heart remodeling (Figure 1). There were linear correlations among PTGES-2 molecular expression in EAT and the diameters (diastolic and systolic), volume (EDV) and mass (LVM and LVM/BSA); this suggests that PGE_2_ biosynthesis in EAT of overweight CVD people is involved in maladaptive cardiac responses.

### 2.4. EP3 Receptor Molecular Expression in EAT Correlates with Body Fatness of Overweight CVD People

EP3 expression correlated substantially with body fatness and waist circumference (Spearman r = 0.43, *p* = 0.05) and WHR (Spearman r = 0.44, *p* = 0.04); and also with factors predicting insulin resistance, such as triglycerides (Spearman r = 0.46, *p* = 0.04) and fasting glucose (Spearman r = 0.50, *p* = 0.03). This suggests that EP3 molecular expression in EAT is related to the increase of body fatness in overweight CVD subjects (Figure 2).

### 2.5. EP3, EP4, and PTGES-2 Are Involved Differently in cAMP Production in EAT

Since PGE_2_ drives both adipogenesis and lipolysis in visceral adipose tissue, acting on intracellular cAMP production by silencing adenylyl cyclase (ADCY) enzymes through its receptors, we ran a correlational analysis between the PTGES-2, EP3 and 4 receptors and the molecular expression of ADCYs in EAT to clarify their effects on cAMP intracellular concentrations in case of excessive (Table 4). PTGES-2 and EP4 were mostly associated with the increase of intracellular cAMP due to the positive correlations between them and ADCY isoforms, suggesting their pro-lipolytic effect on EAT when fat mass increases. In contrast, EP3 molecular expression in EAT seems to be related to anti-lipolytic signaling due to the inverse associations with the main ADCY isoforms in cAMP production, suggesting a protective role against lipolysis during fat mass increase (Table 4).

### 2.6. The PTGES-2 Gene Correlates Directly with EPAC2 as the Molecular Inducer of the ST2/IL-33 Mechanosensitive System

Considering the role of PGE_2_ in the regulation of cAMP concentrations in visceral adipose tissue, and of EPAC2 cAMP effector in the control of cardiac stretch genes such as ST2 and IL-33, we explored the relations between the expression of PTGES-2 and EPAC2 cAMP effector gene (Figure 3a).

PTGES-2 correlates positively with the local expression of EPAC2 (Spearman r = 0.70, *p* < 0.0001) which was recently recognized as one of the main inducers of ST2 gene in EAT. The local protein production of EPAC2 in EAT biopsy suggests active control of EPAC2 in adipocytes due to the stroma immune-localization of EPAC positive cells (black arrows).

That PTGES-2 is involved in sST2/ST2/IL-33 cardiac stretch mediators is further confirmed by the molecular relations between PTGES-2 and ST2, IL-33 gene expression in EAT (Figure 3b). PTGES-2 directly correlates with ST2 gene (Spearman r = 0.70, *p* < 0.0001) which encodes for both ST2 cardiac stretch mediators (ST2L and sST2) and inversely with IL-33 gene (Spearman r = −0.36, *p* = 0.04), which transducer for the main alarmin in the body able to block the circulating isoform of ST2 gene, promoting cardiac cell survival and preventing fibrosis and heart remodeling.

### 2.7. Increase of EAT Mass Deregulates EP3 and EP4 Molecular Expression with Direct Induction of ST2 Gene via EPAC2 cAMP Effector

In the light of the opposite effects of EP3 and EP4 in cAMP intracellular concentrations and the PGE_2_′s role in the induction of sST2/ST2/IL-33 cardiac stretch mediators in immune cells through EPAC2 cAMP effector, we explored the molecular interaction between EP3 and EP4 and EPAC2 and sST2/ST2/IL-33 mediators in EAT of overweight people (Figure 4). Regarding EP4 receptor, which is closely involved in lipolytic processes, gave a positive relation between EPAC2 cAMP effector (Spearman r = 0.46, *p* = 0.001) and ST2 gene (Spearman r = 0.63, *p* = 0.002), and an inverse relation with IL-33 molecular expression, although close to the statistical significance (Spearman r = −0.36, *p* = 0.05) (Figure 4a). Noteworthy, there is a positive association between EP4 molecular expression and EAT thickness (Spearman r = 0.70, *p* = 0.0003) (Figure 4a). There was an interesting inverse relation between EP3 adipogenic receptor and ST2 gene (Spearman = −0.37, *p* < 0.03) and the soluble protein of ST2 receptor sST2 (Spearman R = −0.60, *p* = 0.008) (Figure 4b) which is a powerful mediator of maladaptive heart remodeling released in response to cardiac overload. The EP3 receptor was inversely related to EPAC2 as cAMP effector and inducer of ST2 gene (Spearman r = −0.47, *p* = 0.006) (Figure 4b).

To verify whether the increased EAT mass in overweight persons influences the local protein expression of PGE_2_ receptors, we used western blot analysis to quantify EP3 and EP4 proteins in EAT (Figure 4c). The EAT of overweight persons produced less EP3 anti-lipolytic receptor than EP4 pro-lipolytic receptor, suggesting that an increase of EAT mass can deregulate PGE_2_ control on lipolytic processes via EP3 reduction, contributing to a local increase in cAMP.

## 3. Discussion

EAT is a transducer in obesity and inflammation due to the release of different adipokines that can influence the metabolism of neighboring tissues, especially of the myocardium, on account its anatomical position [41]. In obesity, EAT changes its biological characteristics, with consequent structural and functional abnormalities, leading to impaired myocardial microcirculation, increased LV volume and size, and LA dilatation [41,42], as in our overweight CVD subjects. The anthropometric parameters of abdominal obesity—BMI and waist circumference—are currently used for predicting CVD events associated with insulin resistance [7]. We observed associations between body fatness and echocardiographic parameters of heart maladaptation. The waist circumference, recognized as the best anthropometric measurement of abdominal fat, directly correlates with EAT thickness and both LV and LA dilatation as predictive factor of maladaptive heart response in overweight CVD patients [43]. The question whether the molecular pattern of EAT can shift to a dysfunctional state when abdominal fat mass increases, with the promotion of maladaptive LV structural changes, could be partly answered by our molecular study, where PGE_2_ molecular regulation is pivotal in the activation of ST2 cardiac stretch mediator via EPAC2 cAMP effector. 

PGE_2_ is a potent lipid mediator secreted by various cell types in visceral adipose tissue and it appears to be implicated in the regulation of inflammation and adipocytes functions. Different studies have noted that the excess accumulation of visceral fat depends on depot-specific expression of key enzymes involved in adipose tissue functions, including PGE_2_ biosynthesis-related enzymes [38]. In obesity, PTGES-2 promotes adipose tissue dysfunction, with sustained inflammation and fibrosis, impaired adaptative thermogenesis and increased lipolysis. PTGES-2, also known as cyclooxygenase-2, has in fact been shown to be linked to the early onset of type 2 diabetes and insulin resistance in chronic low-grade inflammation state, especially through the production of PGE_2_ [38]. In view of the involvement of body fatness in heart metabolism, we explored PTGES-2 expression in EAT from overweight CVD patients and echocardiographic parameters of LV enlargement. Our overweight CVD subjects showed a linear correlation between PTGES-2 molecular expression in EAT and cardiac abnormalities associated with LV remodeling, denoting a possible involvement of dysfunctional EAT metabolism and maladaptive heart response. Since adipogenic and lipolytic processes are driven by intracellular cAMP concentrations under PGE_2_ control, we examined the molecular relation between PGE_2_ metabolism in EAT and the expression of genes linked to adenylyl cyclases. There was a positive association between PTGES-2 and the main isoforms of adenylyl cyclases responsible for cAMP synthesis, underling its involvement in intracellular cAMP concentrations during fat mass increase. This is confirmed by the molecular association of PGE_2_ receptors with ADCYs genes. There was an interesting inverse relation between EP3 receptor and ADCY isoforms, confirming its involvement in adipogenic processes, and noteworthy direct associations between EP4 and ADCYs involved in intracellular cAMP increases. The increase of lipolysis driven by cAMP intracellular level, may be one of the factors contributing to obesity-related insulin resistance controlled by PGE_2_ receptors in visceral adipose tissue. The EP3 receptor leads to lower rates of lipolysis and in adipose tissue its deletion may promote an obese phenotype in adult mice [27]. We found EP3 molecular expression correlated positively with the factor predicting insulin resistance, suggesting its potential protective role against obesity-related disorders in overweight CVD subjects. In view of the role of cAMP intracellular levels in the regulation of adipogenic and lipolytic processes driven by PGE_2_ metabolism, and the pivotal role of PTGE-2 in obesity-related disorders, we investigated the involvement of PTGES-2 in the expression of EPAC2, known as a cAMP effector and recently associated with ST2/IL-33 mechanosensitive system involved in the maladaptive heart response. We previously demonstrated when there is an increase in EAT mass, EPAC2 can upregulate ST2 gene, which is a powerful inducer of both cardiac and fat remodeling [23]. Through alternative splicing this gene can transduce for ST2L and sST2 proteins with opposite biological effects: the transmembrane isoform, ST2L can promote cell survival and anti-fibrotic signaling through binding with the IL-33 alarmin protein in cardiac and fat cells [15,18]. In contrast, the truncated soluble isoform, sST2, functioning as a decoy receptor, sequesters IL-33 alarmin protein into extracellular space, preventing ST2L/IL-33 signaling and promoting cardiac and fat tissue maladaptive responses [18]. In view of the role PTGES-2 in obesity and in the control of intracellular cAMP concentrations through PGE_2_ receptors [30,32], we investigate the possible association between PTGES-2 and EPAC2 cAMP effector and ST2/IL-33 mechanosensitive genes. We found active expression of EPAC2 in EAT from overweight CVD subjects and an interesting direct association between PTGES-2 and EPAC2 genes EAT mass increases. Moreover, PTGES-2 inversely correlates with IL-33 alarmin expression and positively with the ST2 gene, reinforcing the hypothesis that PGE_2_ metabolism, which controls intracellular cAMP levels, can also influence the molecular expression of the ST2/IL-33 mechanosensitive genes, through EPAC2. Since in a murine model of macrophages PGE_2_ played a pivotal role in production of IL-33 through EP4-cAMP-EPAC [39] dependent pathway, we looked into the molecular involvement of EP3 and EP4 expression in EAT from overweight CVD subjects and ST2 and IL-33 expression as molecular transducers of maladaptive tissue response. The EP4 receptor correlated directly with EAT thickness, suggesting that EAT mass increase promotes EP4 local expression. Moreover, EP4 directly correlated with EPAC2 and ST2 genes and inversely with IL-33, suggesting its involvement in maladaptive tissue response through EPAC2/ST2 signaling. In contrast, EP3 seems involved in ST2L local expression in case of EAT mass increase and inversely correlates with total sST2 circulating levels, suggesting its possible role in the prevention of maladaptive tissue responses. The local protein production of EP3 and EP4 receptors is expressed differently in EAT from overweight CVD patients. EP4—implicated in lipolytic processes—are more present than EP3 receptor protein, suggesting that when EAT mass increases, deregulated PGE_2_ metabolism seems to be addressed to increased intracellular cAMP level, with consequent upregulation of EPAC2 cAMP effector which is closely involved in ST2 gene expression.

## 4. Materials and Methods

### 4.1. Study Population

This study is conducted on 33 male CVDs patients enrolled at I.R.C.C.S. Policlinico San Donato (San Donato Milanese, Milan, Italy) who underwent open heart surgery. Patients with recent acute myocardial infarction, malignant disease, prior major abdominal surgery, renal failure, end-stage heart failure (HF) and more than 3% variation in body weight in the previous 3 months were excluded. Demographic, anthropometric and clinical data including age, sex, and family history of hypertension, diabetes and CAD are recorded. In accordance to the preoperative coronary angiographic examination, 23 were ischemic patients with CAD undergoing elective coronary artery bypass grafting surgery and 10 were VHD patients receiving valvular replacement. Before surgery, EAT thickness was evaluated by echocardiography. The study protocol was approved by the local Ethics Committee (ASL Milano Due, protocol number 2516; date: 28 December 2009) and patients gave their written informed consent to the examination protocol, conducted in accordance with the Declaration of Helsinki, as revised in 2013. A flow chart which shows the experimental procedure of the study setting is included as Appendix A. 

### 4.2. Blood Collection and Measurements 

Blood samples were collected after overnight fasting into pyrogen-free tubes with ethylenediaminetetraacetic acid as anticoagulant. Plasma samples were separated after centrifugation at 1000× *g* for 15 min and were stored at −20 °C until analysis. Fasting glucose, glycated hemoglobin (HbA1c), creatinine and N-terminal pro B-type natriuretic peptide (NT-pro BNP) were quantified with commercial kits using Cobas 6000 analyzer (Roche Diagnostics, Milan, Italy). Plasmatic level of sST2 was assayed by enzyme-linked immunosorbent assays (ELISA) (R&D Systems, Minneapolis, MN, USA).

### 4.3. Quantification of EAT and SAT Collection

Pre-surgical EAT quantification was quantified by echocardiography with a 2.5- to 3.5-MHz transducer probe (Vingmed-System Five; General Electric, Horten, Norway). EAT thickness was measured along the free wall of the right ventricle from both parasternal long-and short-axis views as previously reported [13]. This point is where EAT generally shows the major thickness and is measurable more easily [13]. EAT thickness at level of the right ventricle free wall is normally 7 mm in both male and female healthy lean individuals; no clinical cut of value is currently validated. EAT biopsy samples were harvested adjacent to the proximal right coronary artery prior to initiation of cardiopulmonary bypass pumping. For gene expression analysis, EAT biopsies were stored in Allprotect Tissue Reagent (Qiagen, Hilden, Germany) at −20 °C until RNA and protein extraction. For immunohistochemical staining assays, EAT biopsies were immediately fixed in paraformaldehyde 4%. To validate RT-PCR assay we collected subcutaneous fat depot (SAT) as control tissue and SAT biopsies were treated like EAT (Appendix A).

### 4.4. Echocardiography Data of Left Ventricular Mass (LV)

Pre-surgical resting echocardiography (Vingmed-System Five; General Electric, Horten, Norway) was performed to assess systolic, diastolic and valvular morphology and function. LV hypertrophy was defined according the current guidelines for echocardiographic chambers quantification.

The outcome measures were LV diastolic diameter (reference values (RV): male 4.2–5.8 cm), LV systolic diameter (RV: male 2.5–4.0 cm), LV end diastolic volume (EDV) (RV: male 62–150 mL), LV end systolic volume (ESV) (RV: male 21–61 mL), LV ejection fraction (EF) (RV: male 52–75%), septal wall thickness (RV: male 0.6–1.0 cm), relative wall thickness (RWT) (RV: male < 0.42%), LV mass (RV: male 88–224 g), indexed LV (LVM/BSA) (RV: male 49–115 g/m^2^), left atria (LA) (RV: male < 4 cm), tricuspid annular plane systolic excursion (TAPSE) (RV: male > 17 mm) and pulmonary artery pressure (PAP) (RV: male < 35–40 mmHg).

### 4.5. DNA Microarray Chip Array Expression Assay

Total RNA was extracted from EAT biopsies with the RNeasy Lipid Tissue Kit (Qiagen). RNA concentration was quantified by NanoDrop 2000 (ThermoScientific, Wilmington, Germany) and RNA integrity was assessed using the Agilent RNA 6000 Nano kit and the Agilent 2100 Bioanalyzer (Agilent Technologies, Santa Clara, CA, USA). Gene expression analysis was performed by one color microarray platform (Agilent). 50 ng of total RNA was labelled with Cy3 using the Agilent LowInput Quick-Amp Labeling kit-1 color, according to manufacturer’s instructions. RNA was purified with the RNeasy Lipid Tissue Mini Kit (Qiagen) and the amount and labelling efficiency were measured with NanoDrop. Hybridization was performed using Agilent Gene Expression hybridisation Kit and scanning with Agilent G2565CA Microarray Scanner System. Data were processed using Agilent Feature Extraction Software (10.7) with the single-color gene expression protocol and raw data were analyzed with ChipInspector Software (Genomatix, Munich, Germany). In brief, raw data were normalized on single probe level based on the array mean intensities and statistics were calculated based on the SAM algorithm by Tusher. From microarray chip analysis, the gene expression of RAPGEF3 (encoding for EPAC1), RAPGEF4 (encoding for EPAC2), gene associated with PGE2 signaling including PTGES-2 (encoding for PTGES-2 enzyme), PTGER3 and 4 (encoding for EP3 and E4 respectively genes, and remodeling mediators including IL1RL1 (encoding for ST2L and sST2) and IL-33 were evaluated and expressed in arbitrary unit (AU).

### 4.6. Real Time Reverse-Transcription PCR (RT-PCR) Assay

To validate our microarray results, we performed RT-PCR assay for only our target genes. Briefly, total RNA was extracted as previously described in the above section. First strand cDNA was synthesized using RT^2^ first strand kit (Qiagen). Quantitative PCR analysis was then performed with RT^2^ SYBR Green Fast Mastermix (Qiagen) and PCR technology (Rotor Gene Q, Qiagen). Relative quantification of mRNA expression in the gene of interest was calculated using the comparative threshold cycle number and the difference between EAT and SAT was evaluated using 2^−ΔCT^ method. Data were normalized by GAPDH levels and expressed as percentage relative to controls. All PCRs were performed at least in triplicate for each experimental condition. GAPDH (Qiagen, PPH00150F), PTGER3 (Qiagen, PPH01838B), PTGER4 (Qiagen, PPH02677A), PTGES2 (Qiagen, PPH16120A), IL1RL1 encoding for ST2 (Qiagen, PPH01076A), IL-33 (Qiagen, PPH17375E), RAPGEF3 encoding for EPAC1 (Qiagen, PPH02838A) and RAPGEF4 encoding for EPAC2 (Qiagen, PPH10495C) genes were measured.

### 4.7. Western Blot Analysis

EAT tissue was homogenized using Minute^TM^Total Protein Extraction kit for Adipose Tissue/Cultured Adipocytes kit (Invent Biotechnologies, Inc), according to the manufacturer’s instructions. Aliquots of 30 μg of total proteins were electrophoresed on SDS Mini-PROTEAN^®^ TGX^TM^ Stain-Free Precast Gels (BIO RAD, Hercules, CA, USA) and transferred to nitrocellulose membrane of Trans-Blot^®^ Tranfer System Transfer Pack (BIO RAD) on Trans Blot^®^ Turbo^TM^ device (BIO RAD). Membranes first blocked with 5% nonfat dry milk/TBS with 0.1% (Vol/Vol) Tween 20 for 1 h and then incubated overnight at 4 °C with primary antibodies for: vinculin (Cell Signaling, Denver, MA, USA; 1:1000), PTGER3 (Proteintech, Manchester, United Kingdom; dilution 1:500) and PTGER4 (Proteintech, Manchester, United Kingdom, dilution 1:700). After washing, membranes are incubated with appropriate horseradish peroxidase (HRP)-labeled secondary antibodies for 2 h at room temperature. Immunoreactive protein bands were then detected using ECL chemiluminescence kit (BIO RAD) using ChemiDoc MP Imaging System (BIO RAD). Desitometric analyses were performed using Image Lab 5.2.1 software (BIO RAD). Data were normalized on total protein quantity after stain-free blot or ponceau staining and presented as percentage density volume (%). All Western Blots were performed at least in duplicate. The full blots were provided in Appendix A.

### 4.8. EPAC2 Immunohistochemical Staining in EAT Sections

Deparaffinised EAT sections were rehydrated and antigen retrieval was performed by autoclaving in sodium citrate buffer 0.01 M pH 6 for 5 min at 120 °C. After rinsing in PBS 1X, quenching of endogenous peroxidases activity was performed in 0.3% H_2_O_2_ in PBS for 20 min. To block unspecific binding, sections were incubated with normal swine serum (Dako Cytomation) and then with the following primary antibody: mouse monoclonal anti-human EPAC2 (diluted 1:400 in PBS, Cell Signaling) overnight and overnight. Sections were then rinsed in PBS and processed for the amplification of immune signal using anti-mouse HRP-polymer complex (MACH 1 Universal HRP-Polymer detection, Biocare Medical, Concord, CA, USA). BIOCARE’s Betazoid DAB was used for color development Sections were counterstained with Mayer’s hematoxylin and mounted with Mowiol 4–88.

Immunohistochemical reactions were observed with a Nikon Eclipse 80i microscope and images acquired by the digital camera and the image acquisition software.

### 4.9. Statistical Analysis

Data were expressed as mean ± standard (SD) and analyzed by GraphPad Prism 5.0 biochemical statistical package (GraphPad Software, Inc., San Diego, CA, USA). The normality of data distribution was assessed by the Kolmogrov-Smirnoff test. Comparison between groups was performed using two-tailed unpaired Student *t* test or Mann-Whitney U-test as appropriate. Spearman or Pearson correlation analyses were used to examine the association between different variables. All differences with *p* < 0.05 was considered statistically significant.

## 5. Conclusions

In summary, our data reinforce the current knowledge on PGE_2_ control of cAMP levels through EP3 and EP4 receptors, and, in case of EAT mass increase, EP3 deregulation seems to be associated with the increase in lipolytic processes with consequent molecular upregulation of the newer inducer of ST2/IL-33 mechanosensitive system, the EPAC2 cAMP effector, in overweight CVD subjects. Further research is now needed to clarify the role of PGE_2_ metabolism in the induction of ST2/IL-33 so as to pave the way to potential therapeutic strategies to prevent cardiac/fat tissue maladaptation driven by obesity.

## Figures and Tables

**Figure 1 ijms-21-00520-f001:**
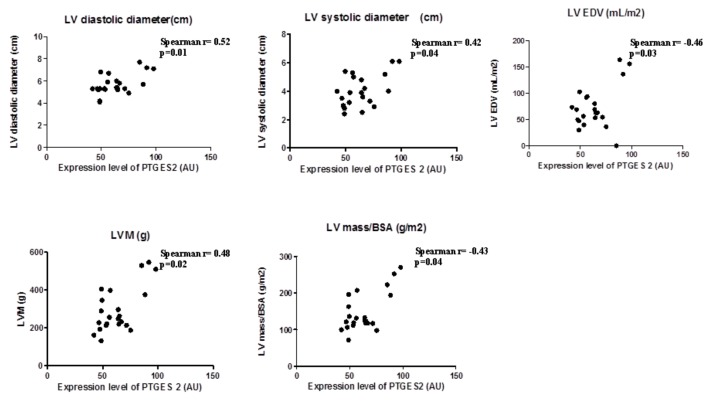
PTGES-2 molecular expression level in epicardial adipose tissue (EAT) of overweight CVD subjects.

**Figure 2 ijms-21-00520-f002:**
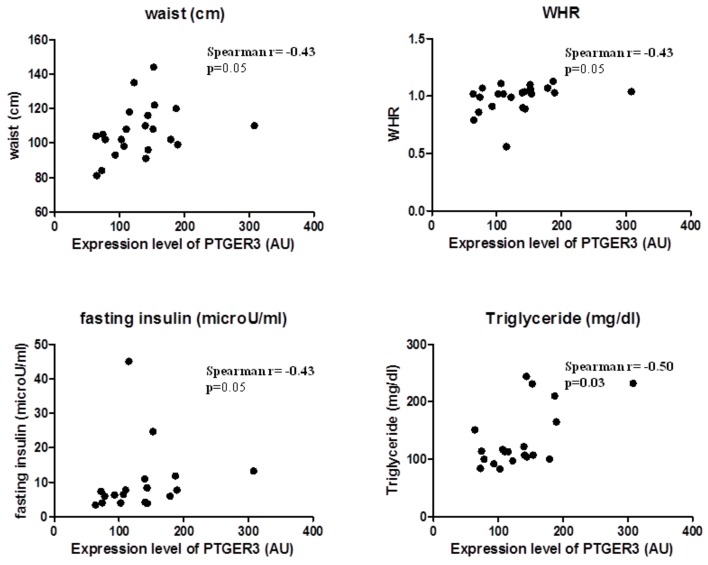
EP3 molecular expression in EAT is associated with body fatness.

**Figure 3 ijms-21-00520-f003:**
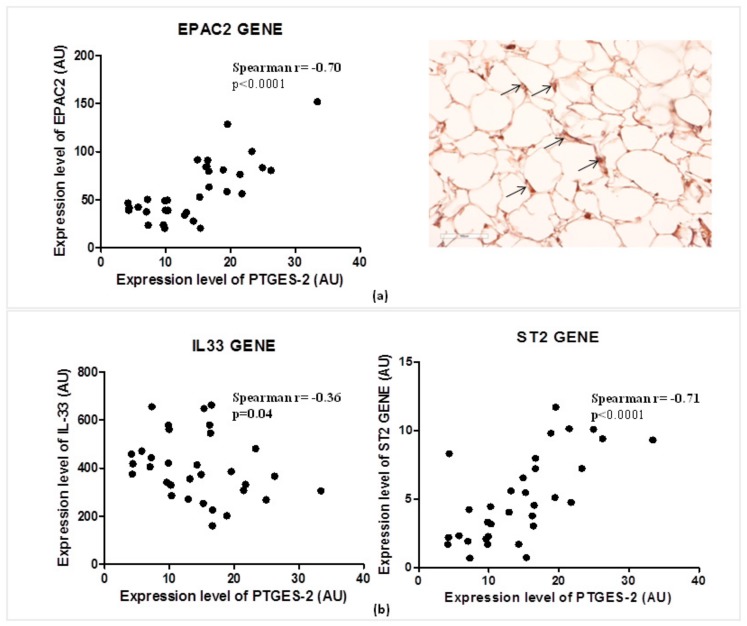
PTGES-2 gene directly correlates with EPAC2 inducer of ST2/IL-33 mechanosensitive system in overweight CVD patients. (**a**) The expression level of PTGES-2 directly correlates with the local expression gene of EPAC2 cAMP effector, powerful inducer of ST2/IL33 mechanosensitive system in immune cells. The local immunolocalization of EPAC2 cAMP effector in EAT biopsies of overweight CVD subjects is demonstrated by EPAC2^+^ cells in the stroma region (black arrows; magnification 20×). (**b**) The PTGES-2 controller of cAMP effectors directly correlates with ST2/IL-33 mechanosensitive genes associated with fat and cardiac maladaptation.

**Figure 4 ijms-21-00520-f004:**
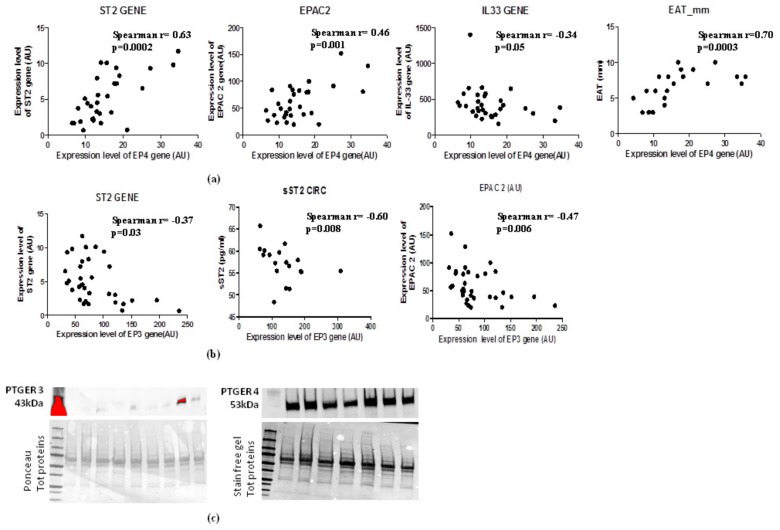
EAT mass increase deregulates EP3 and EP4 molecular expression levels with direct induction of ST2 gene via the EPAC2 cAMP effector. (**a**) EP4 lipolytic PGE_2_ receptor directly correlates with genes associated with mechano-tissue responses including EPAC2 cAMP effector (Spearman r = 0.46, *p* = 0.001) and ST2 gene (Spearman r = 0.63, *p* = 0.002) and inversely with IL-33 molecular expression, although close to the statistical significance (Spearman r = −0.36, *p* = 0.05). (**b**) EP3 anti- lipolytic PGE_2_ receptor correlates inversely with both ST2 isoforms and with EPAC2 cAMP effector. (**c**) EP4 pro-lipolytic isoform of PGE_2_ receptors is more present than the EP3 anti-lipolytic PGE_2_ receptor.

**Table 1 ijms-21-00520-t001:** Cardiovascular disease (CVD) patients’ main details and echocardiographic assessment.

Cardiovascular Patients	Mean	SD	Reference Range
Age (years)	66.86	10.47	/
Systolic blood pressure (mmHg)	130.6	9.66	115–120
Diastolic blood pressure (mmHg)	73.53	4.92	75–80
BMI	27.95	5.19	18.50–24.99
Weight (kg)	83.29	20.48	/
Height (m)	1.71	0.06	/
Waist (cm)	106.70	15.08	<94
Hip (cm)	110.5	25.22	/
HOMA	2.36	2.38	<2.50
WHR	0.98	0.127	<0.95
**Family history**			
Hypertension	3	/	/
Diabetes	2	/	/
CAD	4	/	/
**Biochemical parameters**			
Creatinine (mg/dL)	0.99	0.38	0.60–1.30
Fasting glucose (mg/dL)	46.11	45.04	60–99
HbA1c (%)	4.55	1.43	<6.30
NT-PRO BNP (pg/mL)	453.92	567	<300
Total cholesterol (mg/dL)	155.9	28.86	<200
HDL (mg/dL)	42.48	11.4	40–59
Triglycerides (mg/dL)	132	52.47	<150
Acid uric (mg/dL)	6.64	1.44	4.0–8.0
CRP (mg/100 mL)	0.98	0.38	0.50
ALT (U/L)	28.09	24.67	9.0–60.0
AST (U/L)	32.32	37.33	10.0–40.0
Bilirubin (total) (mg/dL)	0.57	0.31	0.3–1.00

**Table 2 ijms-21-00520-t002:** Echocardiographic assessment of the overweight CVD subjects.

Echocardiographic Data	Mean	SD	Reference Range
EAT thickness in systole (mm)	6.73	2.164	/
**LV internal dimension**			
LV diastolic diameter (cm)	5.68	0.91	4.2–5.8
LV systolic diameter (cm)	3.99	1.13	2.5–4.0
**LV volumes (biplane)**			
LV EDV (mL)	147.5	80.64	62.15
LV ESV (mL)	71.06	49.07	21–61
**LV volumes normalized by BSA**			
LV EDV (mL/m^2^)	75.97	37.04	34–74
LV ESV (mL/m^2^)	35.71	24.25	11.31
**LV EF function**			
LV EF (%)	55.65	11.38	52–72
**LV mass by 2D method**			
septal wall thickness (cm)	1.22	0.2	0.60–1.00
RWT (%)	0.41	0.11	<0.42
LV mass (g)	294.3	119.5	88–224
LV mass/BSA (g/m^2^)	146.9	53.15	49–115
**LA size**			
LA (cm)	4.2	0.67	<4
**RV function**			
TAPSE (mm)	23.2	5.71	>17
**Pulmonary artery pressure**			
PAP (mmHg)	32.23	13.53	<35–40

**Table 3 ijms-21-00520-t003:** Anthropometric measures of body fatness are associated with maladaptive heart remodeling in overweight CVD subjects.

**BMI (x)**	**Spearman r**	***p* Value**
LV diastolic diameter (cm)	0.48	0.02
LV systolic diameter (cm)	0.47	0.03
LV EDV (mL)	0.48	0.03
LV ESV (mL)	0.47	0.04
LVM (g)	0.40	0.05
LA (cm)	0.53	0.03
**Insulin resistance predicting factors**		
Waist (cm)	0.70	0.0004
Fasting insulin (microU/mL)	0.62	0.005
HOMA	0.53	0.02
HDL (mg/dL)	−0.43	0.05
**Waist (x)**	**Spearman r**	***p* value**
EAT thickness in systole (mm)	0.48	0.02
LV diastolic diameter (cm)	0.45	0.03
LVM (g)	0.45	0.03
LA (cm)	0.60	0.01
**Insulin resistance predicting factors**		
BMI	0.70	0.0004
Fasting glucose (mg/dL)	0.46	0.03
Fasting insulin (microU/mL)	0.58	0.001
Triglycerides (mg/dL)	0.43	0.04

**Table 4 ijms-21-00520-t004:** EP3, EP4, and PTGES-2 are involved differently in cAMP production in EAT.

PTGES-2	Spearman r	*p* Value	EP3	Spearman r	*p* Value	EP4	Spearman r	*p* Value
ADCY1	0.73	<0.0001	ADCY1	−0.37	0.04	ADCY1	0.47	0.01
ADCY2	0.66	<0.0001	ADCY2	−0.45	0.01	ADCY2	0.51	0.002
ADCY3	0.04	0.84	ADCY3	−0.34	0.05	ADCY3	−0.19	0.27
ADCY4	−0.13	0.47	ADCY4	−0.11	0.56	ADCY4	0.02	0.89
ADCY5	0.58	0.0005	ADCY5	−0.40	0.02	ADCY5	0.43	0.01
ADCY6	−0.41	0.02	ADCY6	0.62	0.0001	ADCY6	−0.42	0.01
ADCY7	−0.39	0.02	ADCY7	−0.64	<0.0001	ADCY7	0.24	0.16
ADCY8	0.74	<0.0001	ADCY8	−0.46	0.01	ADCY8	0.53	0.001
ADCY9	0.55	0.001	ADCY9	−0.54	0.0011	ADCY9	0.11	0.52
ADCY10	0.78	<0.0001	ADCY10	−0.36	0.04	ADCY10	0.45	0.01

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
