# Peer review of "Correlative Study on Impaired Prostaglandin E2 Regulation in Epicardial Adipose Tissue and Its Role in Maladaptive Cardiac Remodeling via EPAC2 and ST2 Signaling in Overweight Cardiovascular Disease Subjects"

_ijms, 2020, doi:10.3390/ijms21020520_

Round 1
Reviewer 1 Report
The authors have addressed most of the concerns raised by this reviewer. The only issue remaining is the English. Furthermore, the Western blot shown in Fig 4C is saturated (it appears as read in the pdf) which is a bit curious... Dido for supplemental Figure 3 panel B.
Author Response
Point 1: The authors have addressed most of the concerns raised by this reviewer. The only issue remaining is the English. Furthermore, the Western blot shown in Fig 4C is saturated (it appears as read in the pdf) which is a bit curious... Dido for supplemental Figure 3 panel B
Response 1: We would like to thank the Reviewer 1 for your revision. We apologize for our English errors and we improved last version with our English Mother Tongue, hoping that each concern is resolved.
Regarding Western Blot saturation, Fig4C and Fig 3 panel B represent the same membrane. The saturation present on marker lane may be due to non-specific binding of secondary antibody. This normally occurs when the signal in the sample is very low. For this reason, the ECL (very sensitive) can give aspecific signal for overexposure.
Reviewer 2 Report
The authors of the manuscript have addressed my comments in detail.
Author Response
Response 1: We thank you the reviewer 2 for your positive comments.
This manuscript is a resubmission of an earlier submission. The following is a list of the peer review reports and author responses from that submission.
Round 1
Reviewer 1 Report
The manuscript from Vianello et al., provides some insight on a potential correlation between impaired prostaglandin E2 (PEG2) expression in epicardial adipose tissue (EAT) and maladaptive cardiac remodeling, a process allegedly mediated by EPAC2 and ST2 signaling in 33 overweight cardiovascular diseases (CVD) individuals. The study is of interest due to its human nature and advances our knowledge on this topic. However, the authors do not demonstrate the direct effect of PEG2 changes to the observed phenotype. Thus the title should be changed to reflect that it is only a correlative association. Furthermore, the manuscript needs extensive English editing and reorganization. Indeed, some Figure legends include result description which should not be there, while the Discussion requires streamlining to remove introductory material (for example lines 302 to 310 should be in the intro). In addition, a few concerns need to be addressed.
The authors should maintain the definition of the abbreviation EAT as epicardial adipose tissue throughout the text. Indeed, in several instances EAT is define different i.e. line 118 epicardial fat thickness (EAT thickness)? Please provide a reference at line 119. Figures 3 and 4: How can a gene correlate with something? Transcript levels rather than genes were assessed. Legends and graph should reflect Expression levels of PTGSE-2, etc. Figures 3 and 4: From the description provided in ‘Materials and Methods’ it is unclear what approach was used to measure expression levels of each transcript. Was it a DNA microarray chip or a microarray of only the genes of interests? Either way, more details should be provided on the source. Furthermore, as this technique relies on hybridization, all expression levels should be confirmed by quantitative PCR, this is mandatory. Figure 3: A higher quality image of EPAC2 staining should be provided. Figure 4c: Western blots should include a control such as beta-actin and each lane should be labeled. Please provide an image of full blots in a supplementary figure.
Reviewer 2 Report
I appreciate the opportunity to review this interesting paper by Vianello et al. on role of impaired prostaglandin E2 regulation in Epicardial Adipose Tissue (EAT) in overweight CVD patients. It deals with a very timely topic – influence of pathologically changed EAT on maladaptive cardiac remodeling. Authors have found between correlations prostaglandin-endoperoxide synthase 2 (PTGES-2) molecular expression in EAT and the left ventricle (LV) hypethrophy. Another interesting observation was that EP3 receptor expression was correlated with indicators of visceral adiposities such as waist circumference and waist-hip ratio (WHR) measurements. Some their results also suggest possible link of PGE2 deregulation with consequent promotion of Exchange Protein directly Activated by cAMP isform 2 (EPAC2); and ST2 signalling.
Considering these strengths, however the study raises some critical comments.
Abstract
The abstract should be written in a more clearly way. Abbreviations should be introduced at first appearance in the text. Aims of study should be clearly stated. Additionally, the authors’ conclusion in abstract that body fatness increase can shift EAT transcriptome to a pro-tissue remodelling, is an overstatement. They does not have (for obvious reasons) the control group to compare. so and no specific conclusion about cause and effect can be drawn.
Introduction.
Research questions and study objectives are not sufficiently clearly stated in introduction. However authors discussed them in Results part. This information should be moved to the introduction
Results. The authors decided to treat the Results and Discuss as separate parts. However, in some parts it is not clear whether the authors write about their results or other authors. Additionally in several statements there are no references cited. In this part, authors should limit themselves to discussing their own results and discussing the results of other authors should move to the Introduction and/or Discussion.
Discussion. Short conclusions at the end of the discussion would be helpful.
Methods. A flow chart showing the experimental procedure should be included as it makes the study setting more visible to the reader.
Anthropometric measurements are not sufficient for this type of research. If authors wantedwant to show a relationship between the observed changes and obesity, they should use more accurate methods of testing body composition. Especially that they do not have a control group to which they could refer and the study is conducted on a relatively small group.
The authors are encouraged to consult an English editor for professional editing, as there are many typos or incorrectness in grammar, as for example “..adipose tissue including EAT depot[27], which regulates energy metabolism and, in obesity crucially, contributes to fat-inflammation and obesity-associated insulin resistance, through the activation of prostaglandin-endoperoxide synthase 2 (PTGES-2)[30], knowing also as cyclooxygenase 2. PGE2 regulates adipose functions and exerts its biological effects”. It should be rather “known”
